# Antibacterial consumption before, during, and after the COVID-19 pandemic in a tertiary care pediatric hospital in Mexico

Ana Mayra Pérez-Morales[1], Rodolfo Norberto Jiménez-Juárez[2],
Olga Magdala Morales-Ríos[3], Alfonso Reyes-López📶[4]*, Patricia Clark[5],
Erika Janet Islas-Ortega[6], Fernando Ortega-Riosvelasco[7], Heriberto Gómez-Gaytán[8],
María del Carmen Castellanos-Cruz[9], Jessica Liliana Vargas-Neri📶[1]¤*,
the Group of Collaboration ASP-HIM¶

1 Department of Epidemiological Research, Hospital Infantil de México Federico Gómez, Mexico City, Mexico, 2 Department of Pediatric Infectious Diseases, Hospital Infantil de México Federico Gómez, Mexico City, Mexico, 3 Evaluation of Drug and Pharmacovigilance Research Unit, Hospital Infantil de México Federico Gómez, Mexico City, Mexico, 4 Center for Health Economic and Social Studies, Hospital Infantil de México Federico Gómez, Mexico City, Mexico, 5 Clinical Epidemiology Research Unit-Faculty of Medicine, Universidad Nacional Autónoma de México, Mexico City, Mexico, 6 Pharmaceutical Services, Hospital Infantil de México Federico Gómez, Mexico City, Mexico, 7 Department of Hospital Epidemiology, Hospital Infantil de México Federico Gómez, Mexico City, Mexico, 8 Quality Department, Hospital Infantil de México Federico Gómez, Mexico City, Mexico, 9 Central Laboratory of Bacteriology, Hospital Infantil de México Federico Gómez, Mexico City, Mexico

¶ Members of the Group of Collaboration ASP-HIM is provided in the Acknowledgements.
¤ Current address: Department of Epidemiological Research, Hospital Infantil de Mexico Federico Gómez, Cuauhtémoc, Mexico City, Mexico.
* jessivargas4@hotmail.com, jvargas@himfg.edu.mx (JLVN); alfonso.reyes.lopez@outlook.com (ARL)

## Abstract

### Background

During the coronavirus disease 2019 (COVID-19) pandemic, pediatric inpatients frequently received at least one antibiotic, and several antimicrobial stewardship programs (ASPs) strategies were disrupted. In Mexico, no published data are available on antibacterial consumption in children. The aim of this study was to evaluate antibacterial consumption levels and trends before, during, and after the COVID-19 pandemic in a tertiary care pediatric hospital in Mexico, and to forecast future antibacterial consumption.

### Methods

This was a secondary analysis based on time series data of monthly antibacterial consumption from January 2016 to June 2024 in a tertiary care pediatric hospital. Antibacterial consumption was retrospectively measured for the pre-pandemic, pandemic, and post-pandemic periods. Consumption was expressed as days of therapy (DOT) per 1000 patient-days (pd).

**Data availability statement:** All relevant data are within the manuscript and its Supporting information files.

**Funding:** Initial of the author who received an unrestricted award: PC. Grant number awarded: 55159619. Full name of the founder: Pfizer Global Medical Grants. URL: https://www.pfizer.com/about/programs-policies/grants. The founder had no role in study design, data collection and analysis, decision to publish, or preparation of the manuscript.

**Competing interests:** The authors have declared that no competing interests exist.

## Results

The mean antibacterial consumption at the Hospital Infantil de México Federico Gómez (HIMFG) from 2016 to 2024 was 789.3 (95% CI, 756.1–822.5) DOT/1000 pd. In the medical pediatric intensive care unit (M-PICU) and surgical pediatric intensive care unit (S-PICU), mean consumption was 1305.3 DOT/1000 pd (95% CI: 1119.1–1491.6) and 1634.5 DOT/1000 pd (95% CI: 1444.2–1824.9), respectively. Before the pandemic, the hospital-wide mean consumption was 848.8 DOT/1000 pd (95% CI: 811.3–886.2); during the pandemic, it was 709.6 DOT/1000 pd (95% CI: 650.3–769.0); and after the pandemic, 799.2 DOT/1000 pd (95% CI: 698.1–900.3). Overall, consumption rates oscillated around the mean, and no patterns were observed.

## Conclusions

The COVID-19 pandemic did not affect trends of antibacterial consumption across the hospital or in the pediatric intensive care unit. Although the prior authorization component of the ASP maintained antibacterial consumption around the mean, the implementation of additional ASP strategies -such as education and persuasive interventions- alongside current restrictive measures may help further optimize antibacterial consumption in pediatric units.

## Introduction

The coronavirus disease 2019 (COVID-19) pandemic affected many aspects of pediatric infectious diseases, including a) antibacterial use, b) antimicrobial stewardship programs (ASPs), and c) antimicrobial resistance (AMR) [1]. Despite the viral etiology of COVID-19, antibacterials were used as a treatment for both children and adults, even when there was a low rate of bacterial coinfection among patients [2]. The inappropriate use of antibacterials is one of the leading causes of AMR worldwide [3]. Before the pandemic, 60% of hospitalized children had been prescribed at least one antibacterial agent [4]. The days of therapy (DOT) per 1000 patient days (pd) varies across different hospital settings. Neonatal intensive care units (NICUs) and pediatric intensive care units (PICUs) are the wards with the highest DOT values, reaching up to 1440 DOT/1000 pd [5], whereas pediatric hospital-wide data revealed values ranging from 481 DOT/1000 pd [6] to 544 DOT/1000 pd [7]. Thus, a key element in combating the misuse and overuse of antimicrobials that lead to AMR is the implementation of ASPs [8]. ASPs have been implemented in children's hospitals, with reports demonstrating their utility [9]. Several methods are known to be effective interventions within hospital-based-ASPs, including prior authorization, prospective audit feedback (PAF), the combination of prior authorization and PAF, didactic education, the implementation of facility-specific clinical practice guidelines [10], and, recently, the handshake stewardship method [7].

Several studies, including systematic reviews, have reported that up to 75% of patients with COVID-19 including 57% of children with COVID-19 and 75% of adults

with comorbidities received antibacterials [11]. The use of antibiotics among COVID-19 patients in high-income countries (HICs) is 58%, whereas it reached 89% in low- and middle-income countries (LMICs) [12]. Nevertheless, some studies have shown that during the COVID-19 pandemic, the use of antibacterials decreased in pediatric primary care units [13–15], pediatric outpatient clinics [16], and in pediatric community clinics [17,18].

In Mexico, before the COVID-19 pandemic, more than 50% of adult and pediatric inpatients received at least one antibiotic, mainly in public healthcare establishments [19–21]. During the early stages of the COVID-19 pandemic, the proportion of adult patients with severe COVID-19 receiving antibiotics increased to 92% [22]. In addition, there are alarming data concerning the high frequency of empiric antibiotic treatment during the COVID-19 pandemic [22,23] and the increased resistance to third-generation cephalosporins and carbapenems [24]. In Mexico, information regarding antibacterial consumption is presented as the defined daily dose (DDD) per 100 bed-days; however, this metric does not account for the pediatric population. The recommended standard unit of measurement of antibacterial consumption in hospitals is the DDD [25]. However, owing to the variability in body weight, the DDD is not a suitable metric for pediatric populations; instead, DOT is recommended [26]. Early childhood exposure to antibiotics can lead to several gastrointestinal, immunologic, and neurocognitive conditions [27], as well as increased mortality related to AMR.

Pediatric ASPs in Latin America are scarce, possibly because most are still under development or are being incorporated into new standard-of-care strategies [28]. In Mexico, there is a lack of information regarding antibacterial consumption in children before, during and after the COVID-19 pandemic. Understanding patterns of antibacterial consumption in the pediatric population is relevant for improving or implementing ASPs in pediatric hospitals. Therefore, this study aimed to evaluate antibacterial consumption levels and trends before, during and after the COVID-19 pandemic in a tertiary care pediatric hospital in Mexico, and to predict future antibacterial consumption rates.

## Methods

### Setting

The Hospital Infantil de México Federico Gómez (HIMFG) is one of the most important pediatric hospitals in Mexico. It is a tertiary care pediatric hospital with 236 beds distributed across the following wards: medical assistance (cardiology, endocrinology, hemato-oncology, nephrology, pediatrics, infectious diseases, pneumology, rheumatology, neurology, gastroenterology, and neonatology departments), surgical assistance (cardiovascular surgery, thoracic surgery, general surgery, plastic surgery, stomatology, neurosurgery, ophthalmology, orthopedics, otorhinolaryngology and urology departments), neonatal intensive care unit, medical pediatric intensive care unit (M-PICU), surgical pediatric intensive care unit (S-PICU), and emergency department.

The hospital delivers healthcare to approximately 7,000 patients annually, primarily from low-income families not covered by social security systems [29]. The hospital was adapted to treat children with COVID-19 [30]. Among other changes, scheduled surgeries were canceled; however, healthcare delivery for immunocompromised patients (transplant and cancer patients) continued without interruption. In the M-PICU, the entire ward (ten beds) was assigned to patients with COVID-19, while the S-PICU was used for critical patients without COVID-19. From the beginning of the pandemic in Mexico until May 2023, 2134 children were diagnosed with SARS-CoV-2 infection at the HIMFG, of whom 650 were hospitalized; 207 patients required intensive care, 139 patients required mechanical ventilation, and 23 deaths were recorded. All these patients had severe comorbidities.

### Background and rationale of antibacterial use at the HIMFG

In the mid-1980s, hospital authorities initiated the prior authorization stewardship method for prescribing antibacterials, managed by the Infectious Diseases Department (IDD). Initially, this method was implemented for a small group of antibacterials; over time, a broader range was included. Currently, local hospital policy requires prior authorization by the IDD for the use of

carbapenems, third- and fourth-generation cephalosporins, vancomycin, colistin, linezolid, piperacillin/tazobactam, systemic antifungals, posaconazole, antiretrovirals, and systemic antivirals. As part of this stewardship method, pediatricians in the emergency room may prescribe any antibacterial agent as first-line therapy to prioritize timely antimicrobial treatment for severe infections. This indication must then be re-evaluated by the IDD to confirm or adjust the initial treatment.

## Study design

This study was a secondary analysis involving the retrospective analysis of hospital administrative data collected for internal monitoring purposes (Pharmacy and Biostatistics Department databases). Specifically, we analyzed time series data on monthly intravenous antibacterial consumption in a tertiary care pediatric hospital, covering the period from 01/01/2016 to 30/06/2024. Data were accessed between 01/01/2024 to 05/07/2024. The original data collection was conducted independently of the present study and aimed at internal monitoring of antimicrobial consumption and hospital operational metrics. For this secondary analysis, we re-examined these data over approximately nine years to evaluate trends in antibacterial consumption before, during, and after the COVID-19 pandemic. The pre-COVID-19 pandemic phase spanned from January 2016 to February 2020. The COVID-19 pandemic phase spanned from March 2020 to May 2023. The post-pandemic phase spanned from June 2023 to June 2024.

Data on intravenous antibacterial consumption were extracted from the Pharmacy Department. The pharmacy system used for data collection remained unchanged throughout the study period, ensuring data consistency. Data on length of stay were obtained from the Biostatistics Department. All data were aggregated monthly within each phase. The intravenous antibacterials considered for this study included amikacin, amoxicillin, ampicillin, cefepime, cefixime, ceftazidime, ceftazidime/avibactam, cefuroxime, ceftriaxone, cefotaxime, ciprofloxacin, clarithromycin, clindamycin, colistin, dicloxacillin, ertapenem, gentamicin, linezolid, meropenem, metronidazole, penicillin G sodium, piperacillin/tazobactam, tigecycline, trimethoprim/sulfamethoxazole, and vancomycin. No new antibacterials were incorporated during the study period. Although ceftazidime-avibactam was authorized in Mexico for general use in 2015 and for pediatric use in 2019, in our hospital it was only prescribed in April 2024, and therefore its consumption was minimal.

Antibacterial consumption was measured in days of therapy (DOT) per 1000 patient days (pd). DOT was used because it is the recommended measure for measuring drug consumption in the pediatric population. The DOT metric minimizes the impact of the dose variability, unlike the defined daily dose (DDD), which is commonly used in the adult population [31,32]. The DOT is calculated as follows:

$$DOT = (\text{antibacterial days})/(\text{total hospital days during a period}) * 1000$$

When patients received an antibacterial regimen that combined several drugs, the DOT value increased regardless of the magnitude of antibacterial activity. For example, if one patient received two antibacterials for 7 days, this was counted as 14 DOT (7 for antibacterial 1 and 7 for antibacterial 2) [33]. These calculations were performed for both hospital-wide data and for data specific to the PICU. Initially, we analyzed hospital-wide data, followed by subgroup analyses of the S-PICU and M-PICU separately.

## Data analysis

All patients who received at least one systemic antibacterial agent during hospitalization between January 2016 and June 2024 were included. The data were initially processed in Excel and included patient-level information (ID, name, and department) and treatment details (antibiotic, dose, frequency, route, and administration date). Exclusion criteria included incomplete records (i.e., missing date, dose, or frequency of administration) and inconsistencies in treatment data (e.g., repeated dates, incorrect doses, or conflicting frequencies). Data were grouped monthly by antibiotic and department, and DOT (days of therapy) was calculated accordingly.

The mean and 95% confidence interval (CI) of antibacterial consumption were calculated for the pre-pandemic, pandemic, and post-pandemic periods. An overall average, regardless of the COVID-19 pandemic, was also considered. The mean DOT values between periods (pre-pandemic, pandemic, and post-pandemic) were compared using one-way ANOVA and Bonferroni and Tukey test were conducted for pairwise comparisons. ANOVA was initially performed to compare the mean DOT between periods.

Data were aggregated on a monthly basis for time series analysis. Trends and level changes in DOT were explored visually using line plots. To identify cyclical patterns, we applied smoothing techniques. We employed simple and double exponential methods, selecting the most appropriate method for each series. Univariate time series analysis was performed following the Box–Jenkins methodology, with the specific aim of forecasting future antibacterial consumption trends [34]. This methodology consists of a 3-stage iterative process: 1) Model identification: Various models were tested based on partial autocorrelation plots and sample correlation plots. The Dickey–Fuller test (ADF) was applied to choose the best model. 2) Model parameter estimation: The selected model was subjected to a White-noise test and Portmanteau test, and only the values with $p > c^2$ were considered suitable for our model. Furthermore, only the models with the lowest Akaike information criterion (AIC) and Bayesian criterion (BIC) values were included in the analysis. 3) Model diagnostics: The final model was tested to assess its forecasting accuracy. All analyses were performed using Stata v.16.0.

## Ethical considerations

The study was part of protocol HIM/2020/010 SSA.1639, which was reviewed and approved by the Research, Research Ethics, and Biosafety Committees of the Hospital Infantil de México Federico Gómez (HIMFG). The Institutional Research Ethics Committee exempted the researchers from obtaining informed consent. No patient identifiers were used, and all data were analyzed in aggregate form.

## Results

The mean antibacterial consumption at HIMFG from 2016 to 2024 was 789.3 (95% CI, 756.1–822.5) DOT/1000 pd (Fig 1). Antibacterial consumption rate oscillated around the mean with no patterns. Before the pandemic, the mean antibacterial consumption oscillated at approximately 848.8 (95% CI, 811.3–886.2) DOT/1000 pd, which decreased to 709.6 (95% CI, 650.3–769.0) DOT/1000 pd during the pandemic and increased to 799.2 (95% CI, 698.1–900.3) DOT/1000 pd during the post pandemic period. Although the comparisons between the three means were statistically significant for overall antibacterial consumption and both types of therapy, these findings alone did not allow us to conclude that the observed changes were independent of time (Table 1).

In terms of trends, in 2016, antibacterial consumption was below the mean, with nearly bimonthly incremental peaks. In 2017, antibacterial consumption remained mostly above the mean. In 2018, a sustained regression toward the mean was observed from February through mid-year, followed by a slight increase. In 2019, we observed a decrease in the consumption trend, falling below the mean at the end of the year. At the beginning of the pandemic, consumption increased; nevertheless, it began to decline during the pandemic, with the lowest rate observed in February 2022. Finally, after the pandemic, the consumption rate was mostly above the mean (Fig 1). No trend pattern was observed after smoothing the time series (Fig 1). Although consumption varied each year, the overall trend was toward the mean. In other words, consumption fluctuated consistently, even during the COVID-19 pandemic. Based on this information, the forecast for hospital-wide antibacterial consumption was estimated (S1 Table). From July 2024 to June 2025, hospital-wide antibacterial consumption is expected to remain around the mean 789.3 DOT/1000 pd (Fig 2).

Regarding service, the M-PICU and S-PICU both experienced considerable increases in overall antibacterial consumption. The mean antibacterial consumption in the M-PICU and S-PICU from 2016 to 2024 was 1305.3 (95% CI, 1119.1–1491.6) DOT/1000 pd and

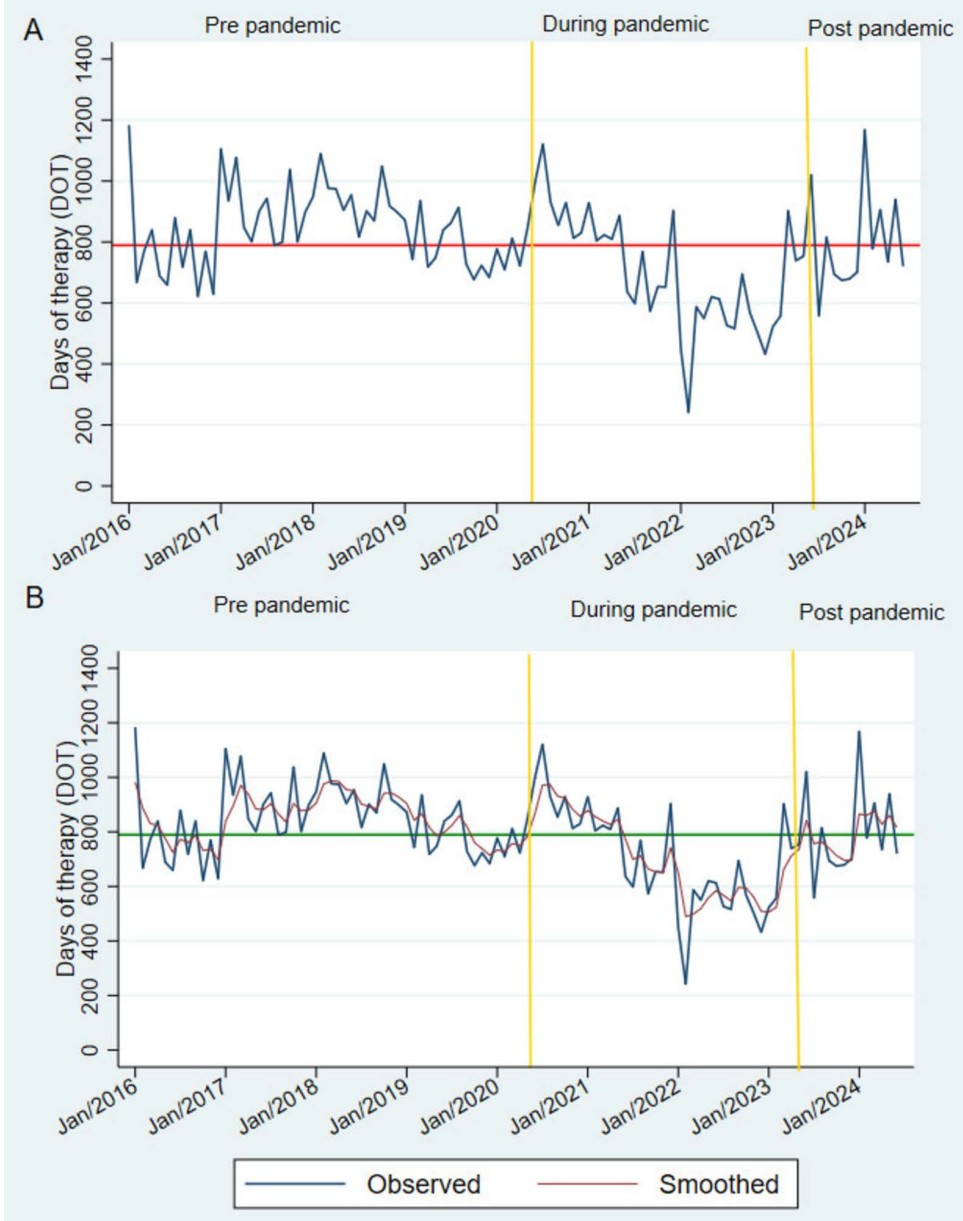

**Fig 1. Line plots of time series data showing antibacterial consumption across all clinical areas of the hospital before and during the COVID-19 pandemic.** Section A shows no trend in the oscillation around the mean DOT (horizontal red line). Section B presents the double exponential smoothing (maroon line) time series.

1634.5 (95% CI 1444.2–1824.9) DOT/1000 pd, respectively (Fig 3). Both units had values above 1200 DOT/1000 pd before and during the pandemic. After the pandemic, both the M-PICU and S-PICU showed decreases in consumption of 38.2% and 45.9%, respectively (Table 1).

The M-PICU showed significant peaks in consumption in mid- to late 2021, with the highest peak observed at 6468.8 DOT/1000 pd. Despite this high level of consumption, consumption rates decreased below the mean after the pandemic. However, the forecast suggests that consumption will slightly increase by 2025 (Fig 4). On the other hand, consumption

**Table 1. Antibacterial consumption before, during, and after the COVID-19 pandemic.**

| | Pre-pandemic antibacterial consumption DOT mean (95% CI) | Pandemic antibacterial consumption DOT mean (95% CI) | Post-pandemic antibacterial consumption DOT mean (95% CI) | *p*-value (F-test) |
|---|---|---|---|---|
| All hospital wards | 848.8 (811.3-886.2) | 709.6 (650.3-769.0) | 799.2 (698.1-900.3) | 0.0004† |
| Medical PICU | 1465.8 (1267.0-1664.6) | 1354.7 (961.0-1748.3) | 560.5 (436.0-685.0) | 0.006* |
| Surgical PICU | 1953.6 (1713.7-2193.5) | 1473.4 (1157.0-1789.9) | 898.6 (648.7-1148.5) | 0.0003** |

Note: The pre-pandemic period was from January 2016 to February 2020 (n = 50 months), the period during the pandemic was considered from March 2020 May 2023 (n = 39 months) and the post pandemic period was considered from June 2023 to June 2024 (n = 13 months).

†The only significant comparison was between the pre-pandemic and pandemic periods (p < 0.001).

*Significant differences were found between the pre-pandemic and post-pandemic periods (p = 0.005), as well as between the pandemic and post-pandemic periods (p = 0.021).

**Significant differences were found between the pre-pandemic and pandemic periods (p = 0.031), and between the pre-pandemic and post-pandemic periods (p < 0.001).

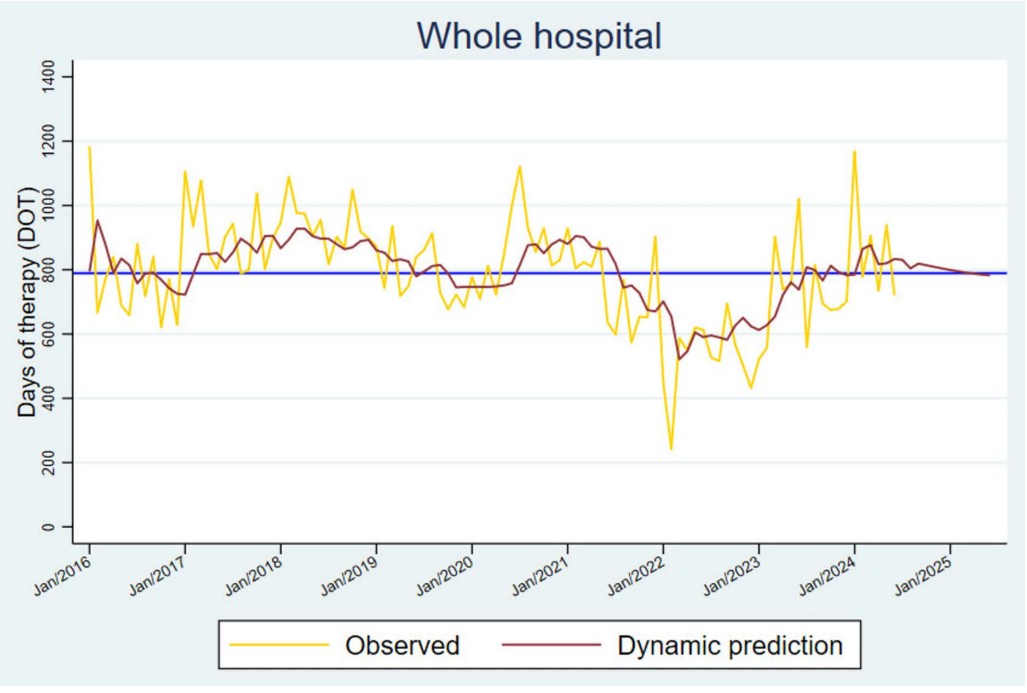

**Fig 2. Time series forecast estimated from the most robust ARMA model for hospital-wide antibacterial consumption.** The dynamic forecast was extended twelve months forward (red line), converging to the mean DOT (blue line).

in the S-PICU remained above the mean both before and during the pandemic. Interestingly, the S-PICU trend showed several peaks in consumption during both periods, with the highest peaks occurring in 2016, late 2020, and March 2022 (3188.9, 3987.5 and 4844.4 DOT/1000 pd, respectively) (Fig 3). The forecast for the S-PICU indicated that antibacterial consumption will increase from July 2024 to June 2025 but remain below the mean (1634.5 DOT) (Fig 4). Smoothing techniques were used for both units, although no cyclicity or trend was observed (Fig 4).

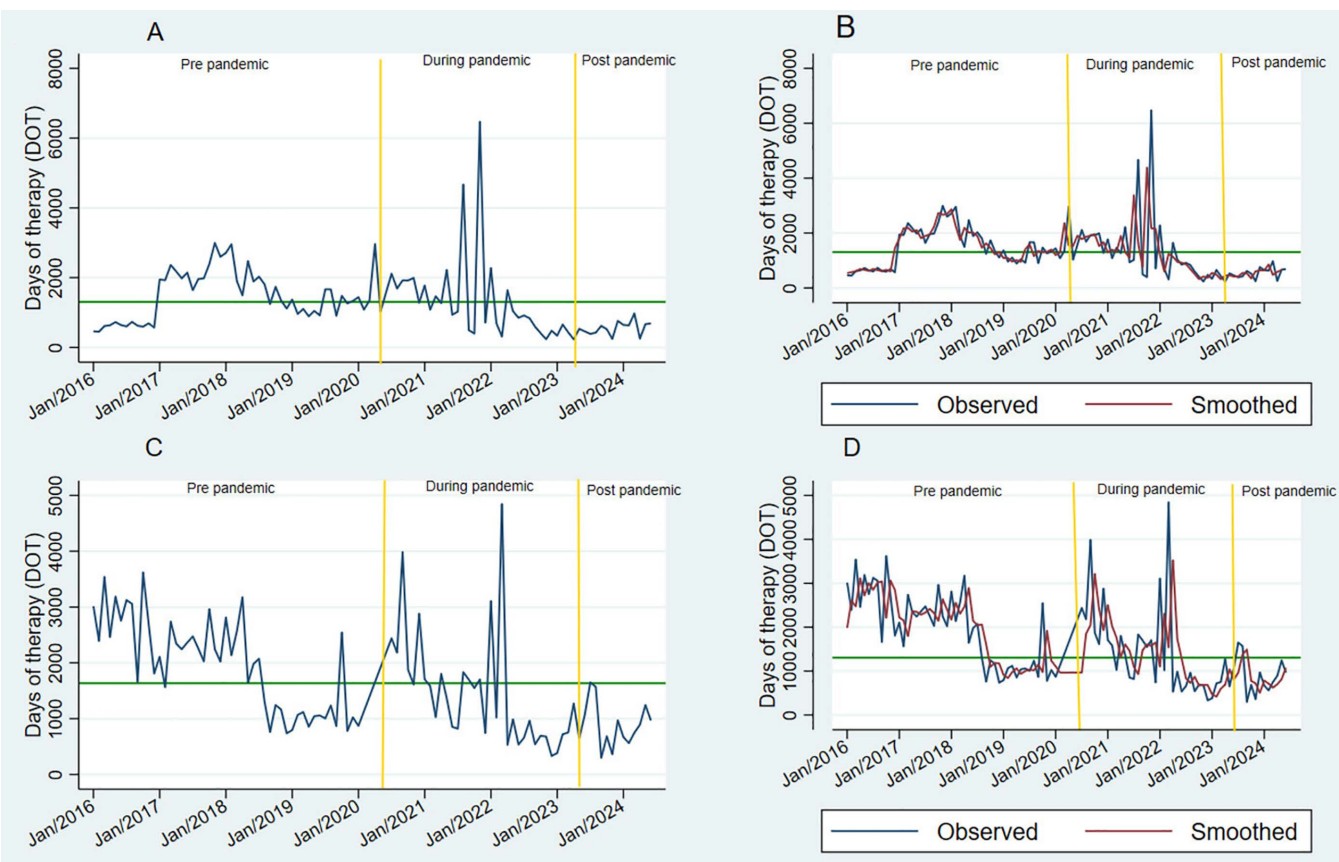

**Fig 3. Line plots of time series highlighting outliers of antibacterial consumption before and during the COVID-19 pandemic.** Sections A and B show no trends in the oscillations around the mean DOT (horizontal red line), corresponding to antibacterial consumption in the PICU (medical and surgical). Sections C and D show double and simple exponential smoothing (red line) for the medical PICU (C) and surgical PICU (D) time series, respectively.

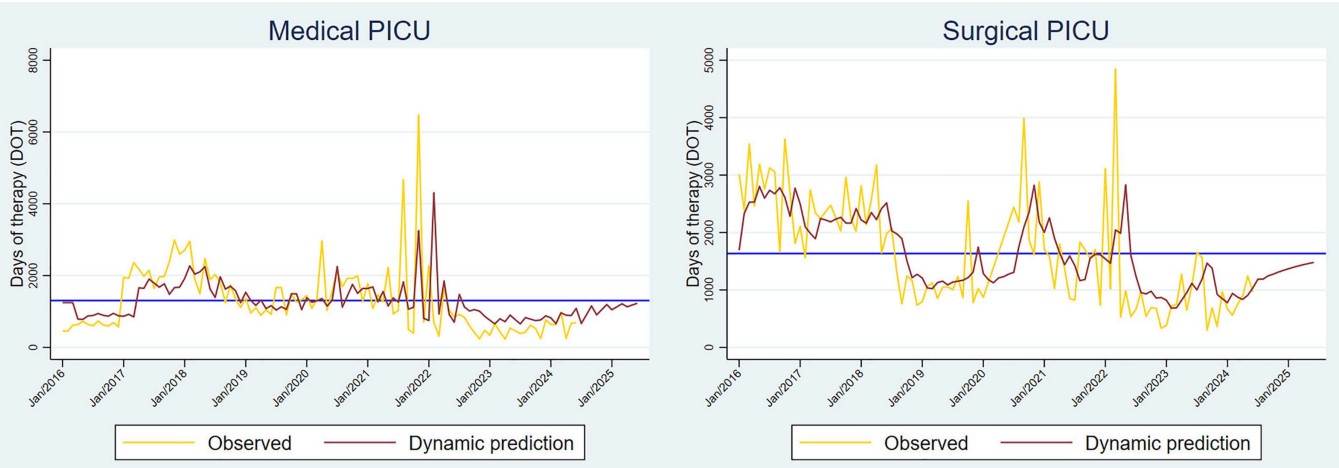

**Fig 4. Time series forecasts estimated from the most robust ARMA model for the M-PICU and S-PICU.** The dynamic forecast was extended twelve months forward (red line), converging to the mean DOT (blue line).

## Discussion

To our knowledge, this is the first study in Mexico to report levels and trends of antibacterial consumption in pediatric patients before, during, and after the COVID-19 pandemic. Additionally, this is the first study to assess pediatric antibacterial consumption using the DOT metric in the country. The mean antibacterial consumption was 848.8 DOT/1000 pd before the pandemic; 709.6 DOT/1000 pd during pandemic, and 799.2 DOT/1000 pd after the pandemic. While the statistical comparison revealed significant differences between the periods evaluated, it is well established that time series data are subject to serial correlation, which can obscure true trends due to the intrinsic variability of the phenomenon under study. Such autocorrelation may produce misleading interpretations when analyzing fluctuations in antibacterial use over time. Therefore, we applied a more robust approach – specifically, the Box & Jenkins methodology – which allowed us to model the temporal behavior of the series. This method enabled us to isolate white noise and generate reliable forecasts of antibacterial consumption trends. The application of this technique not only strengthened the internal validity of our findings but also provided a more comprehensive understanding of the impact of the COVID-19 pandemic on antibacterial consumption over time. It is important to clarify that the ANOVA initially conducted should not be understood as a preliminary step of the Box-Jenkins methodology that we applied later, as both techniques serve very different purposes and were useful for exploring different aspects of our data. While ANOVA provided evidence of differences between means, the Box-Jenkins method allowed us to develop forecasting models.

Over a period of approximately nine years, the hospital-wide antibacterial consumption rate remained around the mean. However, ICUs showed high DOT values. The mean DOT in our hospital was higher than that reported by MacBrayne et al., which was 544 (95% CI 525–562) DOT/1000 pd. They reported that antibacterial consumption decreased after the implementation of handshake stewardship [7]. Unlike the prior authorization method used at HIMFG, handshake stewardship relies on daily in-person feedback from a pharmacist-physician team [7]. Educational and persuasive approaches to modifying antibacterial use have shown to be more effective than restrictive methods. However, due to limited human and financial resources, ASPs in pediatric hospitals in LMICs are scarce and often in early stages of implementation [28]. Prior authorization is recommended as a starting point for ASPs, especially in settings with insufficient resources to support other optimization strategies [35].

Our study revealed that the implementation of at least a prior authorization system helped stabilize antibacterial consumption, even during the COVID-19 pandemic, which forced the restructuring of healthcare systems, including pediatric hospitals [30,36]. However, prior authorization alone is insufficient to modify prescribing behaviors or promote reductions in antibacterial use [37].

Unexpected disruptions caused by the COVID-19 pandemic affected ongoing ASPs in many hospitals [38,39], leading to increased antibacterial use and potentially contributing to rising antimicrobial resistance [40]. In contrast, pediatric outpatient settings saw decreased antibacterial consumption due to fewer respiratory clinic visits [41].

At HIMFG, the prior authorization method remained unchanged during the pandemic. Nevertheless, restrictive interventions can delay treatment and negatively impact the professional culture, especially when communication and trust between infectious disease specialists and clinical teams deteriorate [42]. Interventions such as prospective audit and feedback have shown to be more effective in changing prescribing patterns, promoting prudent and informed antibacterial use [7].

In Mexico, the COVID-19 pandemic had a catastrophic impact, especially among the adult and elderly populations. However, severe cases of COVID-19 in children were rare [43]. In tertiary care hospitals such as HIMFG, pediatric COVID-19 patients were admitted mainly to avoid complications related to underlying diseases, such as cancer. Children with moderate to severe COVID-19 were admitted to the M-PICU. The mean antibacterial consumption in the M-PICU and S-PICU was 1305.3 DOT/1000 pd (95% CI: 1119.1–1491.6) and 1634.5 DOT/1000 pd (95% CI: 1444.2–1824.9), respectively. After the pandemic, both M-PICU and S-PICU experienced a decrease in antibacterial consumption. This may have been due to the resumption of elective surgeries, fewer COVID-19 admissions, increased detection of viral infections

through molecular biology techniques, and greater use of biomarkers in critically ill patients, enabling earlier and more targeted antibacterial treatment.

Antibacterial consumption in our ICUs was comparable to that reported by other LMICs. Araujo Da Silva et al. documented rates ranging from 888.1 DOT/1000 pd (Germany) to 1440.7 DOT/1000 pd (Brazil) [5]. Koopmans et al. reported a rate of 1323 DOT/1000 pd in a South African hospital [44]. In contrast, hospitals in HICs report significantly lower consumption; for example, Dalton et al. reported 75.7 DOT/100 pd in a Canadian tertiary pediatric center [45].

Higher antibacterial consumption in PICUs in LMICs may be associated with different resistance patterns or, concerningly, the absence of fully established ASPs. During the pandemic, uncertainty about treatment options led to the widespread use of antibiotics among COVID-19 inpatients. Our study shows that the pandemic did not significantly affect the trend of overall hospital or ICU antibacterial consumption. This aligns with Vestesson et al., who reported a slight increase in use during the pandemic (from 801 to 846 DOT), but no clear evidence of changes in prescribing behavior [46]. In contrast, a study from China reported an initial decline in antibacterial consumption followed by a subsequent rise [47].

Although hospital-wide trends remained stable, we observed extremely high DOT/1000 pd values in the S-PICU, peaking at 4844.4. This high rate may reflect the complexity and duration of surgeries, which can make it difficult to differentiate between a systemic inflammatory response and nosocomial infection. Furthermore, prior authorization in surgical units at HIMFG has not yet been fully implemented, contributing to surgical prophylaxis being administered without proper stewardship oversight. Prophylactic antibacterial therapy is routinely used perioperatively to prevent surgical site infections, and variability in compliance can inflate DOT rates. Developing uniform recommendations concerning perioperative antibiotic prophylaxis is difficult [48]. This compliance may lead to an increase in DOT.

At HIMFG, many patients present with comorbidities that may complicate surgical outcomes, leading clinicians to continue antibiotics postoperatively. ASPs play a crucial role in academic hospitals like HIMFG by optimizing antimicrobial use and promoting adherence to surgical prophylaxis protocols. A systematic review reported that ASPs increased adherence to prophylaxis guidelines in 85.7% of studies, and 28.5% also showed reduced surgical site infection rates [49].

Finally, our forecast indicates that future antibacterial consumption at HIMFG, both hospital-wide and in ICUs, will return toward the mean. Therefore, ASP interventions—particularly those focused on healthcare worker education—may be critical to modifying this trend.

This study had several limitations. First, we only collected data on antibacterial consumption, not on other antimicrobial classes such as antifungals or antivirals. Second, we did not assess the appropriateness of prescriptions or patient outcomes, nor did we evaluate the unintended consequences of antimicrobial use. Antibacterial consumption metrics can be influenced by several variables, including infection type, access to antibiotics, and safety monitoring. Third, we analyzed overall antibacterial consumption but did not explore trends for specific agents; thus, we could not identify which antibiotics were most frequently used. Nevertheless, our findings show that a single in-hospital intervention helped stabilize antibacterial consumption during the pandemic. These results can serve as a foundation for the development of a comprehensive ASP at our institution and as a benchmark for pediatric hospitals evaluating antibacterial consumption using the DOT metric.

## Conclusions

The COVID-19 pandemic did not affect the trend of antibacterial consumption either hospital-wide or in the PICUs. Although the prior authorization method within the ASP helped maintain antibacterial consumption around the mean, the overall levels were higher than those reported in pediatric hospitals in high-income countries (HICs), but comparable to those observed in low- and middle-income countries (LMICs). Therefore, the implementation of additional ASP strategies—such as educational and persuasive interventions—alongside the current restrictive approach may further optimize antibacterial use in pediatric patients in the future.

## Supporting information

**S1 Table. Results of the Box-Jenkins second stage comparing parameter estimations for several ARMA models of data from whole hospital.** † selected model for the prognosis *$p < 0.05$; ** $p < 0.01$; *** $p < 0.001$.
(DOCX)

## Acknowledgments

The Group of Collaboration Antimicrobial Stewardship Program of Hospital Infantil de México Federico Gómez (ASP-HIM) (Participants are arranged by Department): Clinical Epidemiology Research Unit: Patricia Clark Peralta, Alma Lidia Almiray Soto. Department of Epidemiological Research: Jessica Liliana Vargas Neri, Ana Mayra Pérez Morales, Dulce Blanco Vega, Bertha Edith Reyes Pérez, Jocelyn Jacobo Mendoza, Aleydis Dyveke Hernández Terrazas. Evaluation of Drug and Pharmacovigilance Research Unit: Olga Magdala Morales Ríos. Center for Health Economic and Social Studies: Alfonso Reyes López. Pediatric Infectious Diseases Department: Rodolfo Jiménez Juárez, Martha Avilés Robles, Karla Ojeda Diezbarroso, Almudena Laris González, Silvieluz George Atriano. Epidemiology Department: Fernando Ortega Riosvelasco, Roberto Moreno Miranda, Ana Estela Gamiño Arroyo, Víctor Eduardo López Moreno, Margarita Torres García, Marisol Medina Pelcastre, Pedro Arturo Mejía Rosales. Quality Department: Heriberto Gómez Gaytán. Nursing Department: Angélica María Hernández Tapia, María Luz Flores. Pharmaceutical Services: Erika Janet Islas Ortega, José Rivero Corona, Gerardo Gabriel Vargas Esquivel. Clinical Laboratory Department: Israel Parra Ortega, María del Carmen Castellanos Cruz, Lilia Pichardo, Raúl Ramírez Mondragón. General Director: Adrián Chávez López. Medical Director: Victor Olivar López. Planning Director: Miriam G. Herrera Segura. Research Director: Mara Medeiros Domingo. Teaching and Academic Development Director: Claudia Gutiérrez Camacho. Medical Assistance Subdirector: Rómulo Erick Rosales Uribe. Former members: Sarbelio Moreno Espinosa, Mónica Villa Guillén, Daniela de la Rosa Zamboni, Vanessa Karina Martínez Lara, Frida Isabel Osnaya Valencia, Xuxek Gpe. Becerril Martínez, Laura Fernanda Oropeza Roldán, Carolina Corres Del Corral, Daniela Vázquez Aldana, Tania Salazar Alva, Maximiliano Resendiz Arrazola, Jonathan Omar Méndez Miranda, Damaris Victoria Merino Frutis. We also acknowledge Berenice Soto-Andrade.

## Author contributions

**Conceptualization:** Rodolfo Norberto Jiménez-Juárez, Olga Magdala Morales-Ríos, Alfonso Reyes-López, Jessica Liliana Vargas-Neri.

**Data curation:** Ana Mayra Pérez-Morales.

**Formal analysis:** Ana Mayra Pérez-Morales, Alfonso Reyes-López.

**Funding acquisition:** Patricia Clark, Jessica Liliana Vargas-Neri.

**Investigation:** Jessica Liliana Vargas-Neri.

**Methodology:** Ana Mayra Pérez-Morales, Alfonso Reyes-López.

**Project administration:** Jessica Liliana Vargas-Neri.

**Resources:** Erika Janet Islas-Ortega.

**Software:** Alfonso Reyes-López.

**Supervision:** Rodolfo Norberto Jiménez-Juárez, Jessica Liliana Vargas-Neri.

**Validation:** Rodolfo Norberto Jiménez-Juárez, Jessica Liliana Vargas-Neri.

**Visualization:** Olga Magdala Morales-Ríos.

**Writing – original draft:** Ana Mayra Pérez-Morales, Rodolfo Norberto Jiménez-Juárez, Olga Magdala Morales-Ríos, Jessica Liliana Vargas-Neri.

**Writing – review & editing:** Ana Mayra Pérez-Morales, Rodolfo Norberto Jiménez-Juárez, Olga Magdala Morales-Ríos, Alfonso Reyes-López, Patricia Clark, Erika Janet Islas-Ortega, Fernando Ortega-Riosvelasco, Heriberto Gómez-Gaytán, María del Carmen Castellanos-Cruz, Jessica Liliana Vargas-Neri.

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
