## [Decision Letter · Decision Letter 0]

16 Mar 2025

Dear Dr. Vargas-Neri,

Thank you for submitting your manuscript to PLOS ONE. After careful consideration, we feel that it has merit but does not fully meet PLOS ONE’s publication criteria as it currently stands. Therefore, we invite you to submit a revised version of the manuscript that addresses the points raised during the review process.

In revising your manuscript, please provide information about the primary analysis and explain why the study is a secondary analysis – this is currently unclear. Please also revise the discussion to address the objectives of the study and revise the English in the manuscript..

We look forward to receiving your revised manuscript.

Kind regards,

Petra Czarniak, PhD

Academic Editor

PLOS ONE

Journal Requirements:

4. Please include captions for your Supporting Information files at the end of your manuscript, and update any in-text citations to match accordingly. Please see our Supporting Information guidelines for more information: http://journals.plos.org/plosone/s/supporting-information .

Reviewers' comments:

Reviewer's Responses to Questions

**Comments to the Author**

1. Is the manuscript technically sound, and do the data support the conclusions?

Reviewer #1: Yes

Reviewer #2: Yes

2. Has the statistical analysis been performed appropriately and rigorously?

Reviewer #1: No

Reviewer #2: Yes

3. Have the authors made all data underlying the findings in their manuscript fully available?

Reviewer #1: Yes

Reviewer #2: Yes

4. Is the manuscript presented in an intelligible fashion and written in standard English?

Reviewer #1: No

Reviewer #2: Yes

Reviewer #1: The study reports the evaluation of antibiotic consumption in a pediatric hospital, before, during, and after the greatest activity of the COVID pandemic.

Information is important and data is of interest. However, there are some observations that the authors could address:

In general:

English needs to be revised, e.g., intensive care unit of neonatology vs neonatal intensive care unit.

There are paragraphs that are not fully understood. E.g., "However, owing to the variability in body weight, the DDD cannot be used directly in the 106 pediatric population as DOT".

Line 78 check PD vs pd.

Methods.

It is not clear what they mean by the study being a secondary analysis. What is the primary analysis?

Although the methodology is widely described (perhaps it would even be advisable to evaluate the possibility of synthesizing some sections), the use of the Box-Jenkins model, whose main objective is to forecast, rather than the comparison between past data series, is not clear.

This may be important, given that at least visually and in total numbers, it would seem that antibiotic consumption decreased during the pandemic period (especially between the end of 2021 and the beginning of 2023; in this regard, it should also be considered that the greatest impact on the pediatric population was not at the beginning of the pandemic (2020), but a little later).

It is possible that there is a higher consumption in this period in the critical units, but a formal statistical comparison does not seem to have been made in this regard, perhaps it should be considered.

It would be worth commenting on whether carbapenems are not used in the hospital in particular, since there is no mention of this group of antibiotics.

Discussion

The discussion seems to have a better order. First, we talk about consumption before and during covid, then we talk about outpatient care in some reports, then we return to covid care in the hospital. Perhaps also consider synthesizing it and focusing it only on the objectives of the study, which is the reporting of antibiotic consumption, and not the discussion of the potential impact of different stewardship strategies. E.g., more or less from line 350 to 380, the discussion does not seem to be related to the study

Reviewer #2: Dear authors. I received the article entitled: Antibacterial consumption before, during, and after the COVID-19 pandemic in a tertiary care pediatric hospital in Mexico for review. Despite the relevance of the manuscript, some points should be clarified as described below

1. Abstract- Please inform a legend for HIMFG

2. Study design- Please describe exclusion criteria. Please inform inform if new antibiotics were incorporated during the study period ( for example, probably ceftazidime/avibactam was not available in 2016) . Please inform if the Pharmacy system was the same during the whole period of if there was some change. There's no mention about ethical aspects

3. Discussion= Please explain the reasons for decreasing antibiotic consumption in PICU and Surgical pediatric intensive care unit after the pandemic conpared with the previous periods

**Do you want your identity to be public for this peer review?** For information about this choice, including consent withdrawal, please see our Privacy Policy

Reviewer #1: No

Reviewer #2: No

---

## [Author Response · Author response to Decision Letter 1]

5 May 2025

Dear Editor and Reviewers,

We thank you for the thorough review of our manuscript and for the insightful comments that have helped us improve the quality and clarity of our work. Below, we provide detailed responses to each comment. All language edits are highlighted in yellow, and specific revisions made in response to reviewer comments are highlighted in red in the “Revised Manuscript with Track Changes” document.

Reviewer #1

The study reports the evaluation of antibiotic consumption in a pediatric hospital, before, during, and after the greatest activity of the COVID pandemic.

Information is important and data is of interest. However, there are some observations that the authors could address:

1. English needs to be revised, e.g., intensive care unit of neonatology vs neonatal intensive care unit.

We appreciate the reviewer’s observation. We have conducted a comprehensive English language revision throughout the manuscript to improve clarity and consistency. For instance, “intensive care unit of neonatology” was corrected to “neonatal intensive care unit,” and other similar phrases were edited accordingly. A native English speaker from a language editing service reviewed the final version. All changes are highlighted in yellow.

2. There are paragraphs that are not fully understood. E.g., "However, owing to the variability in body weight, the DDD cannot be used directly in the 106 pediatric population as DOT".

Line 78 check PD vs pd.

Thank you for pointing this out. We have revised the unclear sentence to improve comprehension. The sentence now reads: “However, owing to the variability in body weight, the DDD is not a suitable metric for pediatric populations; instead, DOT is recommended.” (Page 6, lines 110-112)

Additionally, we have corrected typographical inconsistencies, including the formatting of "pd" (patient days), which now appears consistently throughout the text.

3. It is not clear what they mean by the study being a secondary analysis. What is the primary analysis?

We thank the reviewer for this important observation. We clarified in the Methods section that this study represents a secondary analysis of hospital administrative data collected for internal monitoring purposes (Pharmacy and Biostatistics Department databases). There was no previous publication or predefined primary study. The original data were collected for internal purposes, and this secondary analysis was specifically designed to retrospectively evaluate trends in antibacterial consumption. (Page 8, lines 163-169).

4. Although the methodology is widely described (perhaps it would even be advisable to evaluate the possibility of synthesizing some sections), the use of the Box-Jenkins model, whose main objective is to forecast, rather than the comparison between past data series, is not clear. This may be important, given that at least visually and in total numbers, it would seem that antibiotic consumption decreased during the pandemic period (especially between the end of 2021 and the beginning of 2023; in this regard, it should also be considered that the greatest impact on the pediatric population was not at the beginning of the pandemic (2020), but a little later).

We appreciate this insightful comment. We clarified in the Methods sections that the Box–Jenkins methodology (ARMA models) was used with the specific aim of forecasting future antibacterial consumption trends. (Page 11, line 226)

We confirm that a time series analysis was performed to identify underlying patterns and to evaluate whether external events, such as the COVID-19 pandemic, had a statistically significant impact on the data series. We applied the Box & Jenkins methodology to model the series, using statistical tests to determine whether the behavior was driven by an underlying pattern or was stochastic, characterized by random and time-independent fluctuations. This analysis also enabled the generation of a forecast for antibiotic consumption based on the identified data structure. While the primary goal of the study was to describe and compare antibacterial consumption across pandemic phases, the forecasting analysis was included as an additional exploration to estimate future consumption based on historical patterns.

4.1 It is possible that there is a higher consumption in this period in the critical units, but a formal statistical comparison does not seem to have been made in this regard, perhaps it should be considered.

Thank you for the observation. In response, we have now included statistical comparisons between the periods (pre-pandemic, pandemic, and post-pandemic) using one-way ANOVA for the hospital-wide data and for the PICUs. These results have been added to the Results section (Pages 11,12, lines 251-253 and Table 1), and are briefly discussed in the Discussion section. (Pages 16-17, lines 329-340)

5. It would be worth commenting on whether carbapenems are not used in the hospital in particular, since there is no mention of this group of antibiotics.

We apologize for this oversight. We have updated the “Background and rationale” and “Methods” sections to explicitly mention that carbapenems (meropenem) were included in the analysis. These antibacterials are subject to prior authorization and are actively monitored by the Infectious Diseases Department. (Page 9, line 185)

6. The discussion seems to have a better order. First, we talk about consumption before and during covid, then we talk about outpatient care in some reports, then we return to covid care in the hospital. Perhaps also consider synthesizing it and focusing it only on the objectives of the study, which is the reporting of antibiotic consumption, and not the discussion of the potential impact of different

stewardship strategies. E.g., more or less from line 350 to 380, the discussion does not seem to be related to the study

We thank the reviewer for the suggestion. The Discussion section has been revised and reorganized to focus more directly on the study objectives: the description and analysis of antibacterial consumption trends. Content unrelated to the main outcomes, particularly regarding stewardship strategies not evaluated in this study (lines 350–380), has been reduced.

Reviewer #2

Dear authors. I received the article entitled: Antibacterial consumption before, during, and after the COVID-19 pandemic in a tertiary care pediatric hospital in Mexico for review. Despite the relevance of the manuscript, some points should be clarified as described below:

1. Abstract- Please inform a legend for HIMFG

We have added a legend at the first mention of “HIMFG” in the abstract: “Hospital Infantil de México Federico Gómez (HIMFG)”. (Page 3, line 54)

2. Study design- Please describe exclusion criteria.

We clarified in the Methods section that exclusion criteria included incomplete records (e.g., missing dose, frequency, or administration date) and inconsistent entries (e.g., duplicate records, errors in administration timing). (Page 10, lines 206-211)

3. Please inform if new antibiotics were incorporated during the study period (for example, probably ceftazidime/avibactam was not available in 2016).

We added a statement in the Methods section that none antibacterials were incorporated during the study period. (Page 9, line 187)

4. Please inform if the Pharmacy system was the same during the whole period of if there was some change.

We have clarified that the pharmacy system used to collect antibacterial consumption data remained unchanged throughout the study period, ensuring data consistency. (Page 9, lines 179-180)

5. There's no mention about ethical aspects

We apologize for the omission. We added an Ethical Considerations section stating that the study protocol was reviewed and approved by the institutional Research and Ethics Committee. No patient identifiers were used, and data were analyzed in aggregate. (Page 11, lines 237-242)

6. Discussion. Please explain the reasons for decreasing antibiotic consumption in PICU and Surgical pediatric intensive care unit after the pandemic compared with the previous periods

Thank you for this suggestion. We added a paragraph discussing the likely reasons for decreased antibacterial use in the PICUs after the pandemic, such as the resumption of elective surgeries, fewer COVID-19 admissions, and increased detection of viral infections, leading to more targeted antimicrobial use. (Page 20, lines 409-413)

---

## [Decision Letter · Decision Letter 1]

9 Jun 2025

Dear Dr. Vargas-Neri,

We look forward to receiving your revised manuscript.

Kind regards,

Obed Kwabena Offe Amponsah, PharmD, Ph.D.

Academic Editor

PLOS ONE

Journal Requirements:

Reviewers' comments:

Reviewer's Responses to Questions

**Comments to the Author**

Reviewer #1: (No Response)

Reviewer #2: All comments have been addressed

2. Is the manuscript technically sound, and do the data support the conclusions?

Reviewer #1: Yes

Reviewer #2: Yes

3. Has the statistical analysis been performed appropriately and rigorously?

Reviewer #1: No

Reviewer #2: Yes

4. Have the authors made all data underlying the findings in their manuscript fully available?

Reviewer #1: Yes

Reviewer #2: Yes

5. Is the manuscript presented in an intelligible fashion and written in standard English?

Reviewer #1: Yes

Reviewer #2: Yes

Reviewer #1: Thank you for the opportunity to review the responses to the previous comments to this paper.

The authors address most of the comments. However, there is one point that I think requires further clarification in one of their responses and in the added text (page 17):

1. The authors discuss that “While the statistical comparison revealed significant differences between the periods evaluated, it is well established that time series data are subject to serial correlation, which can obscure true trends due to the intrinsic variability of the phenomenon under study. Such autocorrelation may produce misleading interpretations when analyzing fluctuations in antibacterial use over time. Therefore, we applied a more robust approach—specifically, the Box & Jenkins methodology—which allowed us to model the temporal behavior of the series. This method enabled us to isolate white noise and generate reliable forecasts of antibacterial consumption trends. The application of this technique not only strengthened the internal validity of our findings but also provided a more comprenhensive understanding of the impact of the COVID-19 pandemic on antibacterial consumption over time.”

It seems that the authors suggest that the Box & Jenkins model would be a substitute or complement to the analysis of variance. However, the objectives of the two are very different, as are the results: the former focuses on the differences between groups, and the latter on the development of a prediction model. It is worth clarifying this idea further in the discussion.

There is a typo in that section line 338 (comprenhensive vs comprehensive).

Reviewer #2: Thank you for your review, but I'm not still convinced that CAZ-AVI was available in Mexico since 2016. Could you review this information?

**Do you want your identity to be public for this peer review?** For information about this choice, including consent withdrawal, please see our Privacy Policy

Reviewer #1: No

Reviewer #2: No

---

## [Author Response · Author response to Decision Letter 2]

20 Jun 2025

Dear Editor and Reviewers,

We sincerely thank the reviewers and the editorial team for their thoughtful and constructive feedback, which has helped us further strengthen our manuscript. Below we provide detailed responses to the remaining comments raised.

We would also like to note that all modifications made in response to reviewer comments are clearly highlighted in blue in the Revised Manuscript with Track Changes.

Reviewer #1

Comment 1:

“It seems that the authors suggest that the Box & Jenkins model would be a substitute or complement to the analysis of variance. However, the objectives of the two are very different, as are the results: the former focuses on the differences between groups, and the latter on the development of a prediction model. It is worth clarifying this idea further in the discussion.”

Response:

We agree with the reviewer’s observation and appreciate the suggestion. We have now clarified in the Methods (page 10, line 222) and Discussion section (page 17, line 343) that the use of ANOVA and Box-Jenkins responds to different analytical objectives. Specifically, we have added the following sentence:

“It is important to clarify that the ANOVA initially conducted should not be understood as a preliminary step of the Box-Jenkins methodology that we applied later, as both techniques serve very different purposes and were useful for exploring different aspects of our data. While ANOVA provided evidence of differences between means, the Box-Jenkins method allowed us to develop forecasting models.”

This addition reinforces the conceptual distinction and prevents any misinterpretation regarding the relationship between the two methods.

Comment 2:

“There is a typo in that section line 338 (comprenhensive vs comprehensive).”

Response:

Thank you for pointing this out. We have corrected the typo from “comprenhensive” to “comprehensive” in page 17, line 342.

Reviewer #2

Comment:

“I’m not still convinced that CAZ-AVI was available in Mexico since 2016. Could you review this information?”

Response:

Thank you for this important observation. We have verified the regulatory timeline and clarified the details in the revised manuscript. Ceftazidime-avibactam was approved for use in the general population in Mexico starting in 2015 and subsequently approved for pediatric use in 2019. However, in our specific hospital setting, this antibiotic was only used once, in April 2024, during the study period.

To ensure clarity, we have explicitly stated this in the Methods (page 9, line 187) section of the manuscript, as follows:

“Although ceftazidime-avibactam was authorized in Mexico for general use in 2015 and for pediatric use in 2019, in our hospital it was only prescribed in April 2024, and therefore its consumption was minimal.”

We hope this clarification resolves the concern regarding the availability and usage timeline of CAZ-AVI in our study.

Reference List Review

In accordance with the journal’s requirements, we reviewed the full reference list. No retracted articles were identified, and no changes to the reference list were necessary.

We hope that these clarifications and revisions satisfactorily address the remaining concerns. We thank the reviewers once again for their constructive feedback and are pleased that the manuscript is now considered technically sound and suitable for publication.

---

## [Decision Letter · Decision Letter 2]

14 Jul 2025

Antibacterial consumption before, during, and after the COVID-19 pandemic in a tertiary care pediatric hospital in Mexico

PONE-D-24-50363R2

Dear Dr. Vargas-Neri,

We’re pleased to inform you that your manuscript has been judged scientifically suitable for publication and will be formally accepted for publication once it meets all outstanding technical requirements.

Kind regards,

Obed Kwabena Offe Amponsah, PharmD, Ph.D.

Academic Editor

PLOS ONE

Additional Editor Comments (optional):

Reviewers' comments:

Reviewer's Responses to Questions

**Comments to the Author**

Reviewer #1: All comments have been addressed

Reviewer #2: All comments have been addressed

2. Is the manuscript technically sound, and do the data support the conclusions?

Reviewer #1: (No Response)

Reviewer #2: Yes

3. Has the statistical analysis been performed appropriately and rigorously?

Reviewer #1: (No Response)

Reviewer #2: Yes

4. Have the authors made all data underlying the findings in their manuscript fully available?

Reviewer #1: (No Response)

Reviewer #2: Yes

5. Is the manuscript presented in an intelligible fashion and written in standard English?

Reviewer #1: (No Response)

Reviewer #2: Yes

Reviewer #1: (No Response)

Reviewer #2: Dear authors. All queries were answered. No new comments are necessary. Thank you for your review and feedback

**Do you want your identity to be public for this peer review?** For information about this choice, including consent withdrawal, please see our Privacy Policy

Reviewer #1: No

Reviewer #2: **Yes: ** ANDRÉ RICARDO ARAUJO DA SILVA

---

## [Editor Report · Acceptance letter]

PONE-D-24-50363R2

PLOS ONE

Dear Dr. Vargas-Neri,

I'm pleased to inform you that your manuscript has been deemed suitable for publication in PLOS ONE. Congratulations! Your manuscript is now being handed over to our production team.

Kind regards,

on behalf of

Dr. Obed Kwabena Offe Amponsah

Academic Editor

PLOS ONE